# Sevoflurane Dampens Acute Pulmonary Inflammation via the Adenosine Receptor A2B and Heme Oxygenase-1

**DOI:** 10.3390/cells11071094

**Published:** 2022-03-24

**Authors:** Kristian-Christos Ngamsri, Anika Fuhr, Katharina Schindler, Mariana Simelitidis, Michelle Hagen, Yi Zhang, Jutta Gamper-Tsigaras, Franziska M. Konrad

**Affiliations:** Department of Anesthesiology and Intensive Care Medicine, University Hospital of Tübingen, Hoppe-Seyler-Str. 3, 72070 Tübingen, Germany; kristian.ngamsri@uni-tuebingen.de (K.-C.N.); anika.fuhr@uni-tuebingen.de (A.F.); katharina.schindler@gmail.com (K.S.); simelima@hs-albsig.de (M.S.); michelle.hagen@gmail.com (M.H.); yi.zhang.cn23@gmail.com (Y.Z.); jutta.gamper-tsigaras@uni-klinikum.tuebingen.de (J.G.-T.)

**Keywords:** acute lung injury, neutrophils, anesthetic agent, Adora2b, HO-1

## Abstract

Acute respiratory distress syndrome is a life-threatening disease associated with high mortality. The adenosine receptor A2B (Adora2b) provides anti-inflammatory effects, which are also associated with the intracellular enzyme heme oxygenase-1 (HO-1). Our study determined the mechanism of sevoflurane’s protective properties and investigated the link between sevoflurane and the impact of a functional Adora2b via HO-1 modulation during lipopolysaccharide (LPS)-induced lung injury. We examined the LPS-induced infiltration of polymorphonuclear neutrophils (PMNs) into the lung tissue and protein extravasation in wild-type and Adora2b−/− animals. We generated chimeric animals, to identify the impact of sevoflurane on Adora2b of hematopoietic and non-hematopoietic cells. Sevoflurane decreased the LPS-induced PMN-infiltration and diminished the edema formation in wild-type mice. Reduced PMN counts after sevoflurane treatment were detected only in chimeric mice, which expressed Adora2b exclusively on leukocytes. The Adora2b on hematopoietic and non-hematopoietic cells was required to improve the permeability after sevoflurane inhalation. Further, sevoflurane increased the protective effects of HO-1 modulation on PMN migration and microvascular permeability. These protective effects were abrogated by specific HO-1 inhibition. In conclusion, our data revealed new insights into the protective mechanisms of sevoflurane application during acute pulmonary inflammation and the link between sevoflurane and Adora2b, and HO-1 signaling, respectively.

## 1. Introduction

Acute respiratory distress syndrome (ARDS) is still a major challenge in modern intensive care medicine and is associated with high mortality [1]. ARDS occurs as an immune response due to direct lung tissue damage (e.g., aspiration or lung trauma) or indirectly during systemic inflammation (e.g., sepsis). The immune response during acute pulmonary inflammation is characterized by excessive polymorphonuclear neutrophil (PMN) migration, microvascular leakage, and the release of inflammatory cytokines and chemokines [2]. PMNs are the first cells of the innate immune system, which are recruited from the vessels to the site of infection. Their presence is crucial for the clearance of the pathogen, but an aggravated PMN infiltration into the lung leads to excessive destruction of the tissue and impairs pulmonary dysfunction [3]. Controlling excessive PMN migration in acute inflammatory disorders has been shown to be protective [4,5]. Despite decades of intensive research on pulmonary inflammation, the underlying mechanisms and specific therapeutic strategies remain elusive [6,7].

Patients with ARDS on intensive care units (ICU) require sedation, usually performed by the intravenous anesthetic propofol. In an experimental cecal ligation and puncture-induced peritonitis model, propofol increased morbidity and mortality in rats [8]. Meanwhile, sevoflurane, a volatile anesthetic mainly used in the operation theater, can be applied on ICU units [9,10]. Sevoflurane has been shown to elicit anti-inflammatory effects in hepatic ischemia-reperfusion injury, and in chronic and allergic airway inflammation [11,12,13]. A clinical study demonstrated that sevoflurane led to protective effects by reducing inflammatory cytokines during pulmonary surgery, but the underlying mechanism still remains unclear [14].

Adenosine is a purinergic nucleoside, which is involved in several physiological and inflammatory processes [15,16,17]. During acute inflammation, extracellular adenosine triphosphate (ATP) is released by various cell types [18]. Ectonucleoside triphosphate diphosphohydrolase 1 (ENTPD1, also known as CD39) dephosphorylates ATP to adenosine diphosphate (ADP) and adenosine monophosphate (AMP). Next, 5′-nucleotidase (5′-NT, also known as CD73) hydrolyses AMP to adenosine [19,20,21]. The cellular response by adenosine occurs through the four adenosine receptors A1 (Adora1), Adora2a, Adora2b, and Adora3. Among all adenosine receptor subtypes, Adora2b has the lowest affinity to adenosine and requires high concentrations of the nucleoside to be activated, which is typically achieved in terms of inflammation [22,23,24,25]. Previous studies focused on the impact of Adora2a in acute pulmonary inflammation, but these data could not provide a connection between anesthetic agents and Adora2a [26,27]. Recent works suggested a link between the protective effects of sevoflurane and a functional Adora2b [11,28]. Thereby, Adora2b modulated PMN migration and improved microvascular permeability during acute inflammation [23,24]. Further, adenosine supplementation ameliorated the inflammatory response, highlighting the impact of adenosine and adenosine receptors during acute pulmonary inflammation [29]. Patients on intensive care units have been shown to occasionally have reduced ligand affinity and receptor distribution of adenosine receptors [30].

Volatile anesthetics may directly or indirectly induce the activity of heme oxygenase-1 (HO-1) [31]. HO-1 catalyzes the degradation of heme to biliverdin/bilirubin, ferrous ion, and carbon monoxide [32]. Previous investigations suggested protective effects of HO-1 activation during acute pulmonary inflammation [32,33,34]. Our foregoing study linked these protective effects of HO-1 activation during endotoxin-induced pulmonary inflammation with functional adenosine receptor signaling [35].

The presented work examined the protective effects of sevoflurane on the lipopolysaccharide (LPS)-related PMN migration, microvascular permeability, and release of inflammatory cytokines. Furthermore, we investigated the interaction between the anesthetic agent and Adora2b signaling, and HO-1 modulation, respectively. Our hypothesis was that sevoflurane might influence the expression of the Adora2b receptor and therefore has anti-inflammatory effects in terms of acute lung injury. Patients could then potentially benefit from the therapeutic use of sevoflurane during acute lung injury. Recent literature demands the identification of subgroups of patients for a specific therapy [30].

## 2. Materials and Methods

### 2.1. Animals

We used adenosine receptor A2B gene-deficient mice (Adora2b−/−) (from Katya Ravid D.Sc.; Boston University; School of Medicine; Department of Biochemistry; Boston, MA, USA) and corresponding wild-type animals (C57BL/6J) from Charles River Laboratories (Sulzberg; Germany). Mice were male and between eight- to twelve-weeks old. The Animal Care and Use Committee of the University of Tübingen approved all animal experiments.

### 2.2. LPS-Induced Pulmonary Inflammation and Pharmacological Application

Six to eight animals inhaled nebulized lipopolysaccharide (LPS) (0.5 mg/mL from Salmonella enteritidis; total volume 7 mL; Sigma Aldrich; St. Louis, MO, USA) in a custom-made chamber. LPS exposure initiated typical hallmarks of acute pulmonary inflammation, that is, elevated PMN counts, into the different compartments of the lung and raised microvascular permeability. Further, LPS increased the release of inflammatory cytokines and chemokines. Sevoflurane (2 vol%) was administrated in a custom-made chamber with additional oxygen supply for 30 min. Mice breathed spontaneously without conscious effort. To evaluate the optimal time point for the sevoflurane treatment, time curve experiments were performed. Mice inhaled sevoflurane simultaneously, one hour before and three hours after LPS stimulation. The most dominant effects were observed one hour before LPS administration, so we chose this time point for all subsequent experiments. To evaluate the link between the enzyme HO-1 and the influence of sevoflurane, we modulated the activity of the enzyme by a specific agonist (hemin) or inhibitor protoporphyrin IX (SnPP). Hemin is an iron-containing porphyrin, and a specific HO-1 inductor. Hemin (80 µmol/kg BW; Sigma-Aldrich; USA) was injected 24 h before LPS stimulation intraperitoneally (i.p.). In indicated in vivo experiments, we used the specific HO-1 inhibitor SnPP (50 µmol/kg BW; i.p.; Frontier Scientific; Logan, UT, USA), which was applied 30 min before LPS as described before [36].

### 2.3. In Vivo Migration Assay

To quantify the PMN migration into the different compartments of the lung tissue (intravascular adherent, IV; pulmonary interstitium, IS; bronchoalveolar space, BAL), we used a flow-cytometry based method as described in detail before [37]. An allophycocyanin (APC)-conjugated Gr-1 antibody (clone RB6-8C5; Lymphocyte Culture; Charlottesville, VA, USA) was injected into the tail vein of each mouse to label the endothelial adherent PMNs. In deep anesthesia, we removed all non-adherent leukocytes by flushing phosphate-buffered saline solution without calcium/magnesium (PBS-) through the pulmonary vessels. PMNs from the alveolar space were obtained via bronchoalveolar lavage (BAL). The pulmonary tissue was homogenized and incubated with a fluorescent antibody mix against peridinin-chlorophyll-protein (PerCP)-conjugated CD45 (clone 30-F11; BioLegend; San Diego, CA, USA) and fluorescein (FITC)-labeled Ly6G (clone 1A8; BioLegend; USA). Absolute leukocyte counts were defined in the BAL and lung tissue. Subsequentially, we differentiated between intravascular adherent PMNs (IV; CD45 positive, Ly6G positive, and Gr-1 positive) and interstitial PMNs (IS; CD45 positive, Ly6G positive, and Gr-1 negative), whereas neutrophils from the BAL were labeled separately as described before [37]. For the analyses of the expression of the adhesion molecules on murine PMNs in all three lung compartments, additional staining was performed with CD11a/CD18-Phycoerythrin (PE) (also known as lymphocyte function-associated antigen-1; LFA-1; clone H155-78; 141006; BioLegend; USA), CD162-PE (also known as P-selectin glycoprotein ligand-1; PSGL-1; clone 2PH1; 555306; Becton Dickinson; Franklin Lakes, NJ, USA), CD31-PE (clone MEC 13.3; 553373; Becton Dickinson; USA), and CD44-PE (clone IM7; 103024; BioLegend; USA).

### 2.4. Chemokine Release

The liberation of tumor necrosis factor α (TNFα), interleukin-6 (IL-6), CXCL1, and CXCL2/3 was measured in the BAL of mice three hours after LPS exposure by ELISA (DY410; DY406; DY453; DY452; R&D Systems; Minneapolis, MN, USA) according to the manufacture’s protocol.

### 2.5. Gene Expression

Total RNA was isolated from murine lung tissue by using pegGOLD TriFast (020620-6; Peqlab; Erlangen, Germany), and cDNA synthesis was performed by using a Bio-Rad iScript kit (1706391; Bio-Rad; Munich, Germany) according to the manufacturer’s directions. We evaluated the gene expression of murine intercellular adhesion molecule 1 (ICAM1), vascular cell adhesion molecule 1 (VCAM1), HO-1, CD39, CD73, Adora1, Adora2a, Adora2b, and Adora3 by using the following primers: ICAM1 (5′-CAA TTT CTC ATG CCG CAC AG-3′ and 5′-AGC TGG AAG ATC GAA AGT CCG-3′), VCAM1 (5′-TGA CAA GTC CCC ATC GTT GA-3′ and 5′-ACC TCG CGA CGG CAT AAT T-3′), HO-1 (5′-ATGGCGTCACTTCGTCAGAG-3′ and 5′-GCTGATCTGGGGTTTCCCTC-3′), CD39 (5′-CAG AAA GCC ACC ACA GCA AG-3′ and 5′-GGG TCA GTC CCA CAG CAA TC-3′), CD73 (5′-GTT CTG TCT GTT GGC GGT G-3′ and 5′-GGA TGC CAC CTC CGT TTA C-3′), Adora1 (5′-ATT GTC ACT CAG CTC CCG C-3′ and 5′-TCA CCA GTA CAT TTC CGG GC-3′), Adora2a (5′-TCA ACA GCA ACC TGC AGA AC-3′and 5′-GGC TGA AGA TGG AAC TCT GC-3′), Adora2b (5′-GCA TTA CAG ACC CCC ACC AA-3′ and 5′-TTT ATA CCT GAG CGG GAC GC-3′), and Adora3 (5′-GGG TTC CTG TAC TTC CTC TTG G-3′ and 5′-TCA ACC TCA GCC GCT TAT CC-3′).

To reveal the gene expression of the human ICAM1, VCAM1, and Adora2b we used the following primers and performed RT-PCR: ICAM1 (5′- CGA CTG GAC GAG AGG ATT G-3′ and 5′GAT AGG TTC AGG GAG GCG TG-3′), VCAM1 (5′-GGA GCT GAA TAC CCT CCC AG-3′ and 5′-TTA GGA AAA GAG CCT GTG GTG C-3′), and Adora2b (5′-ATG GGC ACT TTC ACC CTC TG-3′ and 5′-GCT CCG GCA GTA GAT AAG GG-3′). 18s served as housekeeping gene 5′-GTA ACC CGT TGA ACC CCA TT-3′ and 5′-CCA TCC AAT CGG TAG TAG CG-3′.

### 2.6. Microvascular Leakage

As a marker of capillary leakage, we evaluated the Evans blue extravasation in the lung tissue. Six hours after LPS nebulization, Evans blue (20 mg/kg; Sigma Aldrich; St. Louis, MO, USA) was injected into the tail vein. Following this, 30 min later and in deep anesthesia, we removed 500 µL blood from the beating right ventricle and flushed the pulmonary vasculature by an injection of three mL of phosphate buffered solution without calcium/magnesium (PBS-). Lung tissue was removed and homogenized, Evans blue was extracted by formamide, and the final concentration was determined colorimetrically, as described in detail before [38]. Additionally, protein concentration in the BAL was determined according to the standard protocol of the protein assay kit (Pierce^TM^; Thermo Fisher Scientific; Waltham, MA, USA).

### 2.7. Generation of Chimeric Mice

Chimeric mice that express Adora2b either on hematopoietic (PMNs) or nonhematopoietic cells (endothelial and epithelial cells) were generated by transferring bone marrow between wild-type and Adora2b−/− mice as previously described [39]. Briefly, recipient animals were irradiated with 600 rad two times with an interval of four hours in between. Femur and tibia were taken from the donor animal; the bone marrow channel was flushed with hank buffered saline solution with calcium/magnesium (HBSS+). The absolute cell count was determined, and after the second irradiation, the recipient animals obtained 3–5 × 10^6^ cells in 300 μL via the tail vein. Chimeric mice that express Adora2b only on nonhematopoietic cells were received by transplanting bone marrow from Adora2b−/− to wild-type mice (Adora2b+/+); chimeric mice with Adora2b only on hematopoietic cells were obtained by transplanting bone marrow from wild-type (Adora2b+/+) to Adora2b−/− mice.

### 2.8. Immunohistochemistry

PMN detection in the lung tissue was performed by using a Vectastain ABC kit (Vector Laboratories; Linaris; Dossenheim; Germany). Sections were blocked with avidin solution (Vector Laboratories; Burlingame, CA, USA) for one hour to avoid unspecific binding sites and followed by incubation with a rat anti-mouse Ly6G antibody (clone RB6-8C5; ab25377; Abcam; Cambridge, UK). Sections were incubated with biotinylated anti-rabbit IgG (BA-4000; Vector Laboratories; Germany) for one hour, followed by Vectastain ABC reagent (PK-4000; Vector Laboratories; Germany) for 30 min and then incubated with DAB substrate. Nuclear fast red (Linaris; Germany) was used for tissue counterstaining. Rat IgG was utilized as a control (sc-2026; Santa Cruz Biotechnology; Dallas, TX, USA). Tissue slides were processed with a Leitz DM IRB microscope (Leica; Wetzlar, Germany) and analyzed with AxioVision v4.8.2 (Zeiss microscopy; Jena, Germany). PMN counts were enumerated from four random sections in each group by independent observers and reported as number of PMNs per high power field (HPF) as described before [40].

### 2.9. Immunofluorescence

For immunofluorescence staining of the lung tissue, the paraffin-embedded pulmonary sections were fixed for ten minutes in 4% paraformaldehyde. After washing, the tissue sections were permeabilized with 1% Triton X and blocked with 5% BSA in PBS- for one hour. The lung tissue was stained with rabbit anti-Adora2b (sc-7505; Santa Cruz Biotechnology; USA), rabbit anti-HO-1 (BML-HC3001; Enzo Life Sciences; Lörrach; Germany), rabbit polyclonal anti-zonula occludens-1 (ZO-1) (1 mg/mL; Thermo Fisher Scientific; Waltham, MA, USA), mouse monoclonal anti-occludin (0.5 mg/mL; Thermo Fisher Scientific; USA), and goat anti-cytokeratin (sc-17101; Santa Cruz Biotechnology; USA) antibodies. For visualization, the following secondary antibodies were used: goat anti-rabbit IgG Alexa Fluor A488 (A11008; Thermo Fisher Scientific; USA), goat anti-mouse IgG Alexa Fluor A488 (A11001; Thermo Fisher Scientific; USA) and rabbit anti-goat IgG Alexa Fluor 546 (A21085; Thermo Fisher Scientific; USA). For nuclear counterstaining, Roti-Mount FluorCare DAPI (HP20·1; Carl Roth; Germany) was used. Rabbit (sc-2026; Santa Cruz Biotechnology; USA) and mouse IgG (sc-2026; Santa Cruz Biotechnology; USA) served for control immunofluorescence staining (Appendix A). Images were analyzed by using ZEN software (Black edition 2011; Zeiss; Germany) and mean fluorescence intensities were measured by ImageJ (Version 1.49 v; National Institute of Health; Bethesda, MD, USA).

### 2.10. In Vitro Experiments

The transmigration of human PMNs through a human endothelial (HMEC1; ATCC^®^ CRL-3243™; Manassas, VA, USA) and human epithelial (NCI-H441; ATCC^®^ HTB-174; USA) monolayer were performed as described previously [26]. Human endothelial and epithelial cells were cultivated on inserts of a Transwell system (3470; Costar; MA; USA) until reaching confluence. Endothelial cells were seeded on the apical and epithelial cells on the bottom side of the inserts so that cells migrate—comparable to the in vivo migration—from the apical to the basolateral side through the monolayer. Human PMNs were isolated from whole blood samples from healthy volunteers. Cells were treated with continuous sevoflurane vaporization of 2 vol% in a custom-made chamber for 30 min. In the indicated experiments, the Adora2b (PSB 1115; Sigma-Aldrich; Germany) (10 ng/mL) was inhibited. PMN migration through the endothelial or epithelial monolayer was initiated by a neutrophil chemotactic mediator, the macrophage inflammatory protein-2 (MIP-2; 200 ng/mL; #250-15; PeproTech; Germany). After one hour, migrated PMNs in the lower chamber were enumerated by light microscopy.

Further, we evaluated the expression of Adora2a and Adora2b on freshly isolated human PMNs four hours after LPS stimulation or without stimulation and the effects of a sevoflurane treatment by flow cytometry. Therefore, we isolated human PMNs from healthy volunteers and stimulated them with LPS (100 ng; Sigma Aldrich; USA) for four hours. To tackle the Adora2a and Adora2b on PMNs, we used FITC-conjugated Adora2a (sc-32261; Santa Cruz Biotechnologies; USA) and PE-conjugated Adora2b (120319; Novus Biotechnologies; Centennial, CO, USA) specific antibodies.

In additional experiments, we isolated human PMNs and activated them for four hours with LPS (100 ng; Sigma Aldrich; USA) or HBSS+ to quantify the mean fluorescence intensity (MFI) of CD11a/CD18 and CD162 at indicated conditions. Additionally, we determined the effects of sevoflurane (2 vol%; 30 min), the HO-1 inductor hemin (5 µM; Sigma-Aldrich; USA), and the HO-1 inhibitor SnPP (20 µM; Frontier Scientific; USA) treatment on the expression of CD11a/CD18 and CD162 on human PMNs by flow cytometry.

### 2.11. Statistical Analysis and Software

For all the animal experiments, the mice were randomly assigned to the groups, used in alternating sequential order, and experiments were performed at the same time point. Continuous variables were reported as the mean and SEM according to their distribution. The distribution was evaluated by investigating kurtosis, skewness, Q-Q plots, and histograms. Independent-samples *t*-tests were used to compare numerical variables that were approximately normally distributed, while Mann–Whitney tests were used to evaluate skewed variables. For comparisons between more than two groups, one-way analysis of variance was performed and adjusted by Bonferroni correction for multiple comparisons. Residuals were inspected in severe deviation from a normal distribution even after log transformation, Kruskal-Wallis test were used. All statistical tests were two-tailed, and the significance level was set at *p* ≤ 0.05 (corrected for multiple testing). Statistical analysis was performed using GraphPad Prism version 8.4.2 for Windows (GraphPad Software; San Diego, CA, USA).

## 3. Results

### 3.1. Time-Dependent Anti-Inflammatory Effects of Sevoflurane

In preliminary experiments, we determined the optimal time point for the treatment with sevoflurane during acute pulmonary inflammation. The impact of sevoflurane on PMN counts attached to the endothelium (IV), lung interstitium (IS), and in the bronchoalveolar lavage (BAL) was assessed 24 h after LPS exposure by flow-cytometry (Figure 1A). Sevoflurane was applied one hour before (pre) LPS inhalation, immediately after LPS (sim), or three hours (post) after LPS stimulation. Sevoflurane reduced the PMN counts, which were attached to the endothelium, in all three treatment groups almost to baseline levels without inflammation, but only the PMNs were significantly decreased in the pretreatment group. In the lung interstitium, sevoflurane significantly decreased PMN counts at all time points, but the strongest effects were observed with the anesthetic as pretreatment or simultaneous administration. The PMN influx into the BAL was significantly reduced only after sevoflurane pretreatment (Figure 1A). For all subsequent experiments, we chose pretreatment (one hour before LPS stimulation) as the time point for the sevoflurane administration.

Next, we performed gene expression analyses to verify the link between the anesthetic agent sevoflurane and the two adenosine receptors, which both play a crucial role in pulmonary inflammation. Previous studies demonstrated the crucial role of the adenosine receptor A2A during various inflammatory models [27,41,42]. Inflammation dampened the expression of Adora2a and Adora2b in the pulmonary tissue 24 h after LPS exposure. Sevoflurane administration enhanced the Adora2b expression in the lung tissue but showed no effects on the gene expression of Adora2a (Figure 1B). Further, sevoflurane also failed to affect the gene levels of CD39, CD73, Adora1, and Adora3 in the lung tissue 24 h after LPS exposure (Appendix A). The migration of leukocytes, especially PMNs, into the lung tissue represent one inflammatory hallmark in the acute phase of pulmonary inflammation. Based on the alterations of Adora2a and Adora2b gene expression in lung tissue during LPS-induced lung injury, we determined the effects of sevoflurane on the expression of both adenosine receptors on PMNs by flow cytometry. In accordance with our gene expression findings, Adora2a and Adora2b on PMNs were significantly reduced 24 h after LPS exposure. Here, the anesthetic agent enhanced the Adora2b expression on PMNs, but not the expression of Adora2a, confirming our previous data (Figure 1C,D).

To corroborate our in vivo findings, we examined the gene and surface expression of both adenosine receptors on human PMNs. LPS reduced the gene expression of Adora2a and Adora2b in neutrophils four hours after stimulation and sevoflurane reversed the LPS effect only on the gene expression of Adora2b. Adora2a expression was unaffected by the administration of sevoflurane (Figure 1E). In additional experiments, we evaluated the surface expression of Adora2a and Adora2b on human PMNs by flow cytometry. The Adora2a and Adora2b mean fluorescence expressions were reduced four hours after LPS exposure. Sevoflurane administration abolished the effects of LPS stimulation on Adora2b expression and augmented the surface presence on human PMNs. The anesthetic agent failed to improve the Adora2a expression, confirming our in vivo findings (Figure 1F,G).

### 3.2. Protective Effects of Sevoflurane on the Influx of PMNs Depend on a Functional Adora2b

We evaluated the impact of sevoflurane on the LPS-induced PMN accumulation into the lung tissue in wild-type and Adora2b−/− animals by flow cytometry and immunohistochemistry (Figure 2A). Confirming our previous in vivo findings and proofing reproducibility, LPS exposure induced a significant raise in PMN accumulation into all three compartments of the lung in wild-type and Adora2b−/− animals. Sevoflurane treatment resulted in significantly lower PMN counts attached to the endothelium, interstitium, and in the BAL. These protective effects were not observed in Adora2b−/− mice (Figure 2B). To detect the migrated PMNs into the lung tissue and to visualize the effects of sevoflurane on PMN migration, we performed immunohistochemical experiments. PMNs were marked with a specific antibody so that they appear brown. We revealed reduced PMN counts after sevoflurane treatment in wild-type mice and detected no effects in Adora2b−/− animals, confirming our flow cytometry findings (Figure 2C,D).

### 3.3. The Impact of Sevoflurane on Adhesion Molecules during Acute Pulmonary Inflammation

Since sevoflurane ameliorated the PMN migration into the lung tissue and alveolar space, we sought to investigate the impact of the anesthetic agent on various adhesion molecules. Adhesion molecules are associated with cell–cell adhesion and play a critical role in neutrophilic migration during acute inflammation [43]. The investigated adhesion molecules have all been shown to play distinct roles on the different steps of PMN migration through the three compartments of the lung, and the expression patterns change during migration [44].

First, we determined the gene expression of intercellular adhesion molecule 1 (ICAM1) and vascular cell adhesion molecule 1 (VCAM1) in the pulmonary tissue. Both adhesion molecules are expressed predominantly on the endothelium [45]. Twenty-four hours after LPS exposure, we observed a raised ICAM1 and VCAM1 expression in the lung tissue of wild-type and Adora2b−/− animals. Sevoflurane treatment reduced the ICAM1 and VCAM1 expression only in wild-type animals (Figure 3A). The ICAM1 and VCAM1 expression were unaffected by sevoflurane in Adora2b−/− mice (Figure 3B). ICAM1 and VCAM1 serve as a ligand for various integrins, such as CD11a/CD18 (also known as LFA-1) or integrin α4β1, which promote neutrophilic migration [46].

Next, we determined the surface expression of important adhesion molecules such as CD11a/CD18, CD162, CD31, and CD44 on murine PMNs in all three compartments of the lung after LPS stimulation in wild-type animals. Sevoflurane diminished the CD44 expression on PMNs adherent to the endothelium (IV). In contrast, sevoflurane significantly reduced the expression of CD11a/CD18, CD162, CD31, and CD44 in the interstitium (IS) of wild-type animals. The bronchoalveolar lavage (BAL) showed a lower expression of CD11a/CD18 and CD162 on the PMN surface in wild types after the inhalation of sevoflurane (Figure 3C). Therefore, we have detected a crucial effect of sevoflurane on PMNs regarding the expression of adhesion molecules and explaining the reduced PMN migration into all lung compartments.

### 3.4. Protective Effects of Sevoflurane on Microvascular Permeability

Next, we sought to investigate the impact of sevoflurane on LPS-induced protein extravasation, since microvascular permeability is—besides PMN migration—the second hallmark of acute pulmonary inflammation (Figure 4A). Evans blue extravasation was assessed as an indicator for capillary leakage, six hours after LPS exposure [38]. LPS stimulation resulted in a higher protein leakage in the pulmonary tissue of wild-type and Adora2b−/− animals. The sevoflurane administration abolished the LPS-related Evans blue extravasation into the lung tissue of wild-type mice, emphasizing the protective effects of the anesthetic agent (Figure 4B). Sevoflurane failed to reproduce the same protective effects in the Adora2b−/− animals, pointing out the importance of a functional Adora2b for the protective effects of sevoflurane on microvascular permeability (Figure 4C).

Further, we evaluated the impact of sevoflurane on histological changes 24 h after LPS exposure by evaluating the thickness of the alveolar septae in wild-type (Figure 4D) and Adora2b−/− mice (Figure 4E). Sevoflurane significantly reduced the LPS-induced septal thickness in wild-type but not in Adora2b−/− animals. To strengthen our previous findings, we evaluated the protein expression of two important tight junction proteins, zonula occludens-1 (ZO-1) (Appendix A) and occludin (Appendix A), in the lung tissue of wild-type and Adora2b−/− mice by immunofluorescence. ZO-1 and occludin are localized in the pulmonary epithelium and endothelium, respectively, and important for the stabilization of the alveolo-capillary barrier [47]. LPS administration decreased the expression of ZO-1 and occludin in wild-type (Figure 4F) and Adora2b−/− animals (Figure 4G), explaining the raised alveolar-capillary leakage. Sevoflurane recovered the expression of ZO-1 and occludin in the pulmonary tissue only in wild-type animals, highlighting the protective impact of sevoflurane on the alveolar-capillary barrier during acute lung inflammation.

### 3.5. Sevoflurane Dampens the Expression of Inflammatory Mediators

The impact of sevoflurane on the expression of inflammatory mediators in the pulmonary tissue and the BAL was investigated by RT-PCR and ELISA (Figure 5A). We determined the LPS-induced gene expression of two main inflammatory cytokines, TNFα and IL6, in the lung. After LPS exposure, the gene levels of both cytokines were elevated in the pulmonary tissue of wild-type (Figure 5B) and Adora2b−/− animals (Figure 5C). The sevoflurane administration significantly diminished both cytokines only in wild-type animals. Further, we investigated the pulmonary expression of the two main PMN chemoattractants, CXCL1 and CXCL2/3, which resemble the human interleukin 8 (IL8). The gene expression of both chemokines was significantly reduced in wild-type animals after sevoflurane. Confirming our previous findings, sevoflurane had no effects in Adora2b−/− animals in this setting.

Additionally, we quantified the LPS-induced release of TNFα, IL6, CXCL1, and CXCL2/3, into the BAL of wild-type and Adora2b−/− mice and sevoflurane diminished the liberation of all four mediators in wild-type animals (Figure 5D). In contrast, the genetic depletion of Adora2b canceled the protective effects of sevoflurane on the release of all four mediators, confirming our previous findings (Figure 5E).

### 3.6. The Impact of Sevoflurane on Hematopoietic and Non-Hematopoietic Adora2b In Vivo

To verify the effects of sevoflurane on different cell types, we determined the impact of sevoflurane on PMN migration and microvascular permeability in chimeric animals. To generate chimeric mice, which only expressed the Adora2b in the non-hematopoietic cells, we transferred bone marrow from Adora2b−/− animals in wild-type mice (Adora2b+/+). Adora2b−/− mice received bone marrow from wild-type (Adora2b+/+) animals and expressed the Adora2b only on hematopoietic cells (Adorab2-blood) (Figure 6A) [48]. The sevoflurane administration could not affect the PMN influx in chimeric mice, which did not express the Adora2b on hematopoietic cells (Figure 6B). In chimeric mice, which possessed the Adora2b only on hematopoietic cells, the anesthetic induced significantly lower PMN counts in all three compartments, highlighting the crucial role of a functional Adora2b on hematopoietic cells for the protective properties of sevoflurane (Figure 6C).

Evans blue extravasation was also assessed in the generated chimeric mice. LPS exposure induced an elevated Evans blue extravasation into the pulmonary tissue of both generated chimeric animals. Sevoflurane failed to reduce the Evans blue leakage in the lung tissue of chimeric mice, which expressed the Adora2b on hematopoietic cells (Figure 6D), and in the animals, which expressed Adora2b in the pulmonary tissue (Figure 6E). These findings point out the importance of a functional Adora2b for the protective effects of sevoflurane on the maintenance of the microvascular permeability during acute inflammation.

### 3.7. Protective Effects of Sevoflurane In Vitro Depend on a Functional Adora2b

To verify our in vivo data, we performed in vitro experiments with human PMNs and human endothelial and epithelial cells, respectively. The effects of sevoflurane administration and Adora2b inhibition on the migration of human PMNs through a monolayer of human endothelium and epithelium cells were evaluated by using a Transwell system (Figure 7A). The transendothelial (Figure 7B) and transepithelial (Figure 7C) macrophage inflammatory protein-2 (MIP-2)-related PMN migration was significantly reduced by sevoflurane, and the specific pharmacological Adora2b inhibition with PSB1115 abolished this protective effect. The dihydroethidium (DHE) production of fresh isolated PMNs served as a marker for the reactive oxygen species (ROS) activity of human PMNs and was evaluated by flow cytometry. Sevoflurane administration dampened the MIP-2-related DHE activity, and these effects were reversed by specific Adora2b inhibition (Figure 7D). The ICAM1 (Figure 7E) and VCAM1 (Figure 7F) gene expression on HMEC1 were determined four hours after LPS stimulation by RT-PCR. We observed a significantly raised expression of both adhesion molecules after LPS stimulation, while sevoflurane abolished the LPS effects on ICAM1 and VCAM1 expression. Further, we evaluated the LPS-related Adora2b expression on human endothelial cells with or without sevoflurane administration. Sevoflurane abolished the LPS-related effects on Adora2b expression and reversed the Adora2b gene levels to baseline (Figure 7G).

### 3.8. The Impact of the Anesthetic Agent Sevoflurane on HO-1 Modulation during Acute Pulmonary Inflammation

HO-1 is a well-known intracellular enzyme, whose activation provides cyto-protective effects during acute inflammation, hypoxia, and hyperoxia [49]. Previous data suggested a possible link between sevoflurane and HO-1 expression [36,50]. First, we evaluated the impact of hemin and sevoflurane on HO-1 gene and protein expression. HO-1 expression was elevated three hours after LPS exposure in the lung tissue. Sevoflurane, respectively, HO-1 activation with hemin increased the LPS-related HO-1 expression. The simultaneous administration of sevoflurane and hemin enhanced the inflammation-related raise of the HO-1 expression (Figure 8A). Next, we evaluated the protein expression of HO-1 in the pulmonary tissue by immunofluorescence 24 h after LPS exposure. As an inflammatory response, we observed a raised HO-1 expression, which was further elevated by sevoflurane or hemin administration. In line with our gene expression data, simultaneous sevoflurane and hemin application had additive effects on the HO-1 protein expression (Figure 8B and Appendix A). A previously published study suggested a possible connection between volatile anesthetics and the Adora2b through the HO-1 signaling [35]. We evaluated the impact of sevoflurane and HO-1 activation with hemin on Adora2b gene expression by RT-PCR. Both agents, sevoflurane and hemin, raised the Adora2b expression three hours after LPS exposure. The combined treatment with sevoflurane and hemin resulted in a significantly higher Adora2b gene expression compared to a single hemin administration alone (Figure 8C). Further, we examined the effects of sevoflurane and HO-1 modulation on Adora2b surface expression in the lung tissue. LPS administration reduced the Adora2b expression in the lung tissue, which could be reversed by hemin treatment. Sevoflurane enhanced the effects of hemin on Adora2b expression in the lung tissue, highlighting possible additive effects of the anesthetic agent on the expression of Adora2b through HO-1 modulation (Figure 8D and Appendix A).

Next, we evaluated the influence of sevoflurane and HO-1 modulation on the LPS-induced PMN sequestration into the lung tissue. To emphasize the link between sevoflurane and HO-1, we administrated in all following experiments SnPP, a specific HO-1 inhibitor. Hemin ameliorated the LPS-induced PMN influx into the lung tissue. The combined treatment with sevoflurane and hemin led to additive effects with a stronger reduction in PMN counts into the lung interstitium and BAL (Figure 8E). The administration of the HO-1 inhibitor, SnPP, abolished the protective effects of sevoflurane on the LPS-induced PMN migration.

To evaluate our in vivo findings, we performed additional in vitro experiments with human PMNs. We determined the surface expression of CD11a/CD18 and CD162 on human neutrophils, both important adhesion molecules for the migration of human PMNs. LPS administration elevated both adhesion molecules significantly on human PMNs as expected. HO-1 activation with hemin decreased the LPS-related expression of both adhesion molecules. Sevoflurane administration enhanced the protective effects of hemin and abolished the LPS-induced surface expression of CD11a/CD18 (Figure 8F) and CD162 (Figure 8G). HO-1 antagonism suppressed the sevoflurane effects on both adhesion molecules, highlighting the link between sevoflurane and HO-1.

### 3.9. Protective Effects of Combined Treatment with Sevoflurane and Hemin on the Release of Inflammatory Cytokines and Microvascular Permeability

Next, we sought to evaluate the impact of sevoflurane and the combined sevoflurane/hemin treatment on the expression and release of inflammatory meditators three hours after LPS exposure. The anesthetic agent ameliorated the LPS-induced gene expression of TNFα, IL6, CXCL1, and CXCL2/3 almost to the baseline in the pulmonary tissue (Figure 9A). The combined treatment with sevoflurane and hemin showed no additive effects on the expression of all four inflammatory cytokines. The release of TNFα, IL6, CXCL1, and CXCL2/3 into the BAL was assessed by ELISA. Both therapies, sevoflurane alone and the combined treatment, reduced the liberation of all four inflammatory cytokines. The combination, hemin and sevoflurane, showed additive effects on the release of TNFα and IL6 (Figure 9B). SnPP abolished the protective effects of sevoflurane on the release of all four cytokines, indicating the impact of sevoflurane on HO-1 modulation.

Furthermore, we examined the impact of HO-1 activation on microvascular permeability combined with sevoflurane administration. Sevoflurane and hemin significantly reduced the LPS-related protein extravasation in the BAL (Figure 9C). The combined therapy showed no additive effects. Additionally, SnPP cancelled the protective effects of sevoflurane indicating the crucial role of sevoflurane on HO-1 modulation. A schematic overview of the protective effects of sevoflurane during acute inflammation and the interaction with the HO-1 signaling is shown (Figure 9D).

## 4. Discussion

Experimental studies revealed protective effects of sevoflurane in various inflammatory models, such as ischemia-reperfusion liver injury or microbial peritonitis [11,13,51,52,53]. Several clinical trials confirmed the anti-inflammatory properties of anesthetic agents such as sevoflurane during lung resection and cardio surgery, although the underlying mechanisms remain elusive [10,54,55,56]. Our study determined the protective effects of sevoflurane on PMN migration, respectively, inflammation-related adhesion molecules and microvascular permeability during acute pulmonary inflammation. Further, we revealed the link of the anesthetic agent with a functional Adora2b signaling and evaluated the effects of sevoflurane on HO-1 modulation during acute pulmonary inflammation.

Two prospective clinical trials showed that sevoflurane diminished the release of inflammatory cytokines during cardio and lung resection surgery [56,57]. These data were collected in clinical trials with patients in surgery setting, but data from ICUs concerning the effects of sevoflurane in term of acute pulmonary inflammation remain rare. Since the development of anesthetic vaporizers, such as the Anesthetic Conserving Device (AnaConDa^®^), inhaled sedation has become more popular on ICUs worldwide. Jabaudon et al. reported data from a randomized pilot trial, which showed a significantly improved oxygenation and reduced release of inflammatory cytokines in the plasma, respectively, BAL of human patients after sevoflurane sedation [58]. In line with these studies, we observed an ameliorated release of inflammatory cytokines after sevoflurane treatment, and additionally, we determined decreased PMN counts into the inflamed lung tissue and BAL. Numerous experimental studies suggested that the sevoflurane treatment provided protective effects by reducing PMN infiltration in terms of liver and renal ischemia-reperfusion injury, renal injury, and sepsis, confirming our findings [11,28,59,60]. However, these studies missed evaluating the effects of sevoflurane on the surface expression of migration-related adhesion molecules, which are important for the migratory behavior of PMNs. Sevoflurane administration inhibited the expression of migration-related adhesion molecules on PMNs, explaining the reduced PMN sequestration into the lung tissue. In line with our findings, previous data suggested that diminished CD11b expression on leukocytes after sevoflurane administration reduced the tissue injury after ischemia and reperfusion [61].

A hallmark of acute inflammation is a disturbed microvascular permeability, which results in tissue edema [62]. Our study detected an elevated expression of tight junction proteins and improved microvascular permeability after sevoflurane treatment, highlighting the protective properties of the anesthetic agent during the acute inflammatory response. In line with our results, recent studies demonstrated that sevoflurane administration improved the microvascular permeability during cerebral ischemia and lung injury [63,64].

Adenosine is a purinergic nucleoside, which acts via G protein coupled Adora1, Adora2a, Adora2b, and Adora3, and mediates various cellular responses during acute inflammation [65]. In terms of inflammation, extracellular adenosine is rapidly produced through the breakdown of ATP to ADP, and finally to adenosine by the membrane-bound enzymes CD39 and CD73 [65]. Experimental data showed that a reduced extracellular adenosine concentration exacerbated the inflammatory response during acute pulmonary inflammation [66]. On one hand, Kiers et al. observed that elevated extracellular adenosine provided protective effects, such as reduced release of inflammatory cytokines, during endotoxin-related inflammation [67]. On the other hand, adenosine has a short half-life and undesirable side effects, such as atrial fibrillation [68]. However, if administration of adenosine is not feasible, then a modulation of adenosine receptors may be an option. Our data demonstrated that a genetic and pharmacological depletion of Adora2b abrogated the protective effects of sevoflurane, highlighting the impact of Adora2b during acute inflammation. In line with our findings, previously published data pointed out the impact of adenosine receptors during acute sterile and endotoxin inflammation in vivo [69]. Subsequently, experimental studies identified the protective properties of Adora2b activation during ischemia and reperfusion injury and LPS-induced inflammation [22,24,25,48]. These works used BAY60-6583, the most examined Adora2b agonist, and demonstrated the protective effects of Adora2b signaling. To the best of our knowledge, no clinical data with a specific Adora2b agonist are so far available [70]. In our present work, sevoflurane ameliorated LPS-induced PMN migration through a functional Adora2b and elevated the Adora2b expression itself. In line with our findings, previous studies demonstrated the impact of sevoflurane on migrated PMNs, tissue edema, and release of inflammatory chemokines in sterile and polymicrobial peritonitis and peritonitis-related sepsis by inducing the expression of Adora2b [11,28]. Previous experimental studies with mice and rats showed that sevoflurane reduced the PMN infiltration during acute peritoneal inflammation or polymicrobial sepsis through the Adora2b signaling [28,71]. Our studies with chimeric mice indicated that sevoflurane dampened the PMN migration only in the presence of a functional Adora2b on PMNs. To the best of our knowledge, we are the first to show that the protective effects of sevoflurane on the neutrophilic behavior were linked with a functional Adora2b on hematopoietic cells during acute pulmonary inflammation.

Recently, Wang et al. detected that sevoflurane enhanced microvascular permeability during endotoxin-induced lung injury [72]. The colleagues in this study did not evaluate which cell types were responsible for the protective effects of sevoflurane and missed examining the impact of Adora2b on the capillary leakage. We are the first to demonstrate that sevoflurane requires a functional Adora2b on hematopoietic and non-hematopoietic cells to provide protective effects on microvascular permeability. Clinical data from ICU patients suggested that a disturbed adenosine receptor affinity during acute inflammation impaired the immune response, highlighting our findings and emphasizing the impact of functional adenosine receptor signaling [30].

Heme is an iron protoporphyrin-IX and is involved in various peroxidase, monooxygenase, and dioxygenase enzyme activities [73]. Furthermore, heme is involved in various inflammatory disorders by catalyzing pro-oxidant reactions, regulating the inflammatory response, and modulating cell death pathways [73,74]. Previous studies demonstrated that HO-1 modulation provides protective effects during burn injury, acute renal injury, renal ischemia and reperfusion injury, and sepsis [33,75,76,77,78]. Our own previous data demonstrated the protective effects of HO-1 activation on PMN migration, microvascular permeability, and the release of inflammatory cytokines during LPS-induced pulmonary inflammation [36,74]. Lee et al. reported that HO-1 modulation induced the expression of adenosine receptors, Adora2a and Adora2b, respectively, during inflammation, which may serve as an explanation for the protective properties of HO-1 activation [49]. These data are in line with our findings, which showed that HO-1 activation increased the Adora2b expression in the pulmonary tissue, but Lee et al. missed examining the impact of anesthetic agents on Adora2b and HO-1 expression, respectively. Bauer et al. recently suggested a possible link between anesthetic agents such as sevoflurane and the cytoprotective effects of HO-1 modulation during acute inflammation [79]. To the best of our knowledge, we are the first to demonstrate that sevoflurane or hemin, a specific HO-1 activator, enhanced the expression of Adora2b during acute pulmonary inflammation. Moreover, simultaneous sevoflurane administration and HO-1 activation resulted in a synergistic effect concerning a raised Adora2b expression and in an enhanced HO-1 expression itself. The combined sevoflurane and hemin treatment resulted in a significantly reduced PMN migration as well as decreased TNFα and IL6 release. In line with our findings, previous studies demonstrated a link between anesthetic agents and the HO-1 signaling during ventilator-induced lung injury [80,81]. Further, the protective effects of sevoflurane and hemin on PMN migration, microvascular permeability and release of inflammatory cytokines were abolished by SnPP, a specific HO-1 inhibitor, highlighting the impact of the anesthetic agent on HO-1 signaling. Previous data showed that a HO-1 inhibition with SnPP cancelled the anti-inflammatory effects of a specific HO-1 activation during acute inflammation, confirming our results [74].

## 5. Conclusions

The presented study demonstrated that sevoflurane administration abolished the PMN migration into the pulmonary tissue by modulating the expression of various adhesion molecules on PMNs during acute LPS-induced inflammation. Additionally, sevoflurane improved the microvascular permeability and dampened the release of inflammatory cytokines, whereas the protective effects were associated with a functional Adora2b. Next, we demonstrated that sevoflurane enhanced hemin-induced Adora2b and HO-1 overexpression, respectively, and increased the protective effects of HO-1 activation on LPS-related PMN migration, microvascular permeability, and release of inflammatory cytokines. The protective effects of the combined treatment with the anesthetic agent and HO-1 activator were cancelled by specific HO-1 inhibition pointing out the link between sevoflurane and HO-1 signaling (Figure 9D). The present data indicates that sevoflurane may be beneficial for ICU patients during acute pulmonary inflammation.

## Figures and Tables

**Figure 1 cells-11-01094-f001:**
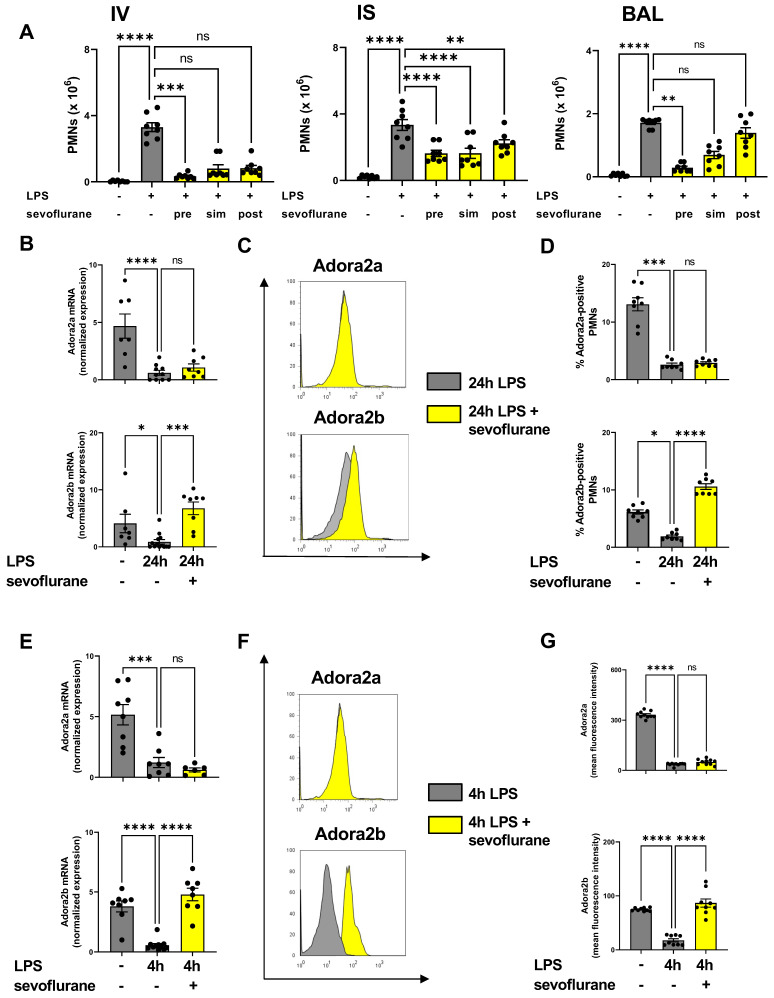
Time-dependency of the anti-inflammatory effects of sevoflurane on PMN migration and the influence of sevoflurane on the expression of adenosine receptors. Sevoflurane was applied one hour before (pre) LPS inhalation, immediately after LPS (sim), or three hours (post) after LPS stimulation. (**A**) Polymorphonuclear neutrophil (PMN) sequestration into the intravascular space (IV), interstitium (IS), and bronchoalveolar lavage (BAL) was assessed in wild-type animals by flow cytometry (*n* = 6–8). (**B**) The gene expression of the adenosine receptors A2A (Adora2a) and A2B (Adora2b) in the pulmonary tissue 24 h after LPS exposure were detected by real-time-polymerase chain reaction (RT-PCR) (*n* = 6–8). (**C**) Adora2a and Adora2b surface expression on migrated PMNs in the pulmonary tissue 24 h after LPS stimulation was evaluated by flow cytometry. (**D**) Percent of Adora2a and Adora2b positive PMNs in pulmonary tissue was evaluated 24 h after LPS exposure (*n* = 8). (**E**) Adora2a and Adora2b gene expression on human neutrophils four hours after LPS exposure and the effects of sevoflurane administration were determined by RT-PCR. (**F**) Representative histogram of Adora2a and Adora2b surface expression on human PMNs after LPS exposure and with sevoflurane administration. (**G**) Mean fluorescence intensity of Adora2a and Adora2b are shown, four hours after LPS stimulation on human PMNs (*n* = 8–12). Multiple group comparison was analyzed by one-way ANOVA and Bonferroni correction or Kruskal-Wallis test. Data are presented as mean ± SEM; ns = not significant; * *p* < 0.05; ** *p* < 0.01; *** *p* < 0.001; **** *p* < 0.0001.

**Figure 2 cells-11-01094-f002:**
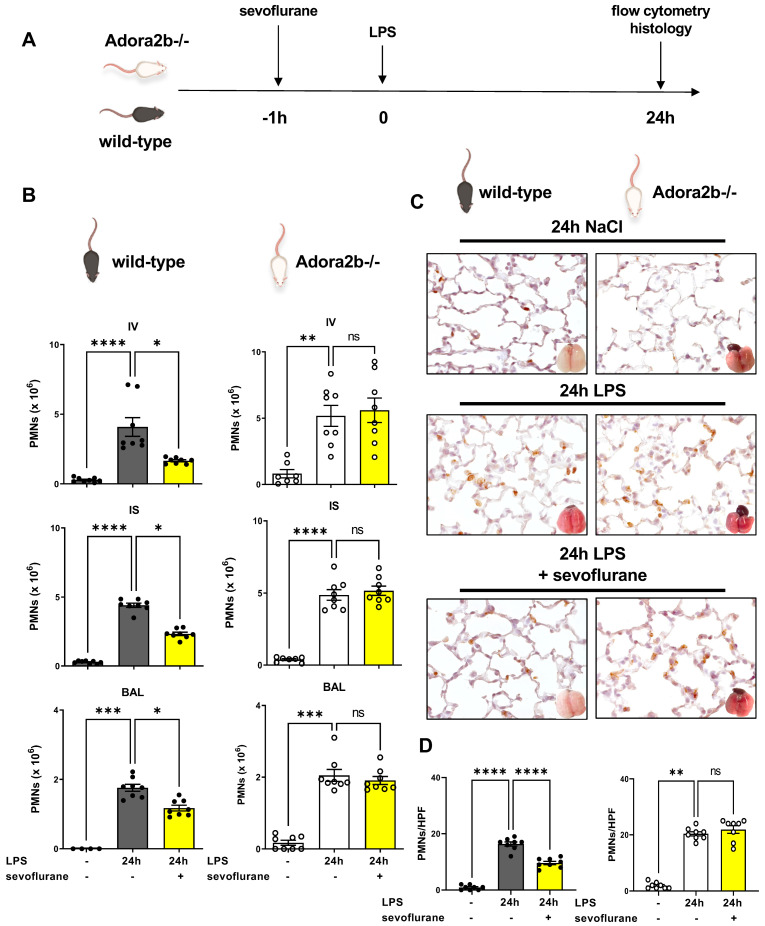
Effects of sevoflurane treatment on lipopolysaccharide (LPS)-induced migration of polymorphonuclear neutrophils (PMNs) in wild-type and Adora2b−/− animals. (**A**) Wild-type and adenosine A2B receptor−/− (Adora2b−/−) mice were treated with sevoflurane. 24 h after LPS exposure, PMNs counts were determined by flow cytometry, and immunochemistry experiments with the pulmonary tissue were performed. (**B**) PMN migration was determined in the intravascular space (IV), interstitial tissue (IS), and the bronchoalveolar lavage (BAL) in wild type and Adora2b−/− mice (*n* = 6–8). (**C**) PMN sequestration into the pulmonary tissue was detected by an immunohistochemical method. PMNs were tackled with specific antibodies so that PMNs appear brown (images are presentative slides of *n* = 3 slides from each animal; *n* = 4 mice per group; 63× magnification). (**D**) PMN counts were enumerated by light microscopy. PMNs from four representative high-power fields (HPF) were counted from three different slides (*n* = 4 mice per group). Multiple group comparison was analyzed by one-way ANOVA and Bonferroni correction or Kruskal-Wallis test. Data are presented as mean ± SEM; ns = not significant; * *p* < 0.05; ** *p* < 0.01; *** *p* < 0.001; **** *p* < 0.0001.

**Figure 3 cells-11-01094-f003:**
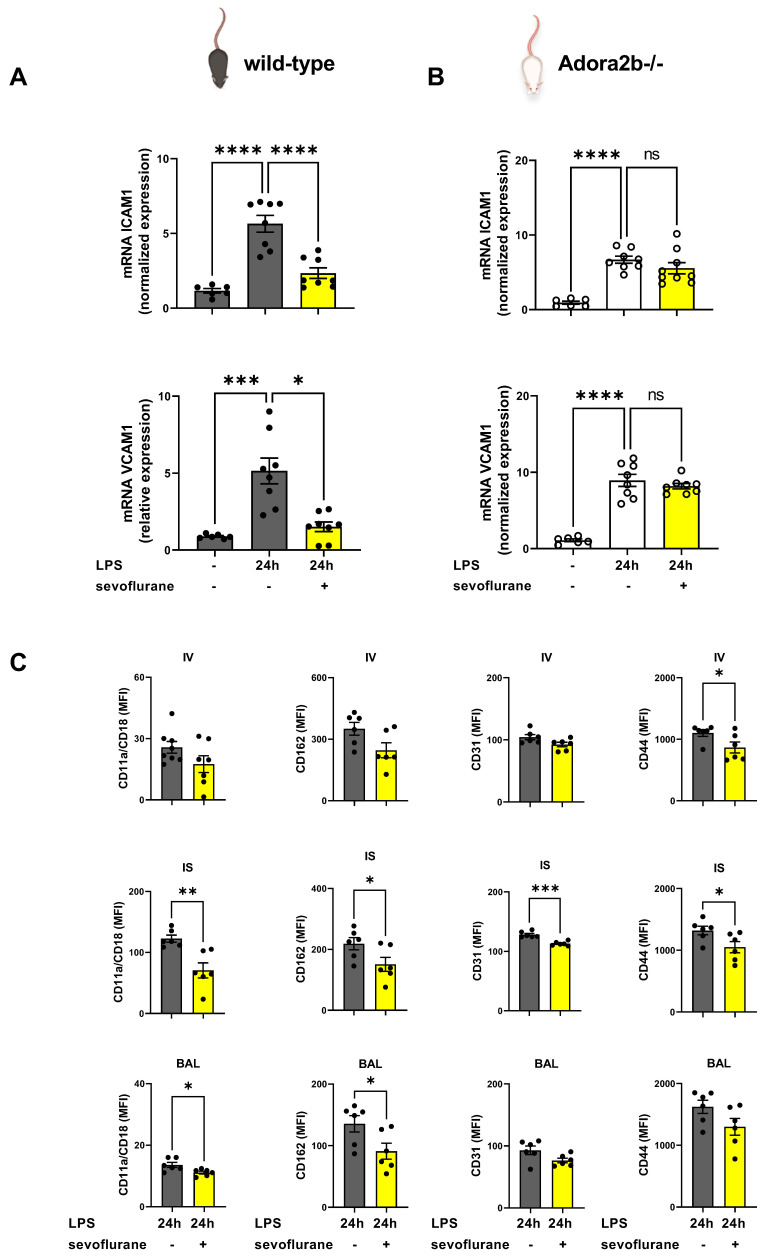
Influence of sevoflurane on the expression of adhesion molecules in the lung tissue and on neutrophils. (**A**) Wild-type, and (**B**) adenosine A2B receptor−/− (Adora2b−/−) mice were treated with lipopolysaccharide (LPS) and sevoflurane, and the gene expression of intercellular adhesion molecule 1 (ICAM1) and vascular cell adhesion molecule 1 (VCAM1) in the pulmonary tissue were assessed by real-time-polymerase chain reaction (RT-PCR) (*n* = 6–8). (**C**) Sevoflurane-induced differences in the mean fluorescence intensity (MFI) of the adhesion molecules, CD11a/CD18 (also known as lymphocyte function-associated antigen 1), CD162, CD31, and CD44, on neutrophils of wild-type animals in the intravascular space (IV), pulmonary interstitium (IS), and bronchoalveolar lavage (BAL) (*n* = 6–8) were assessed by flow cytometry. Statistical analyses were performed by one-way ANOVA + Bonferroni correction or Kruskal-Wallis test. Two-group analyses were derived from unpaired data by *t*-tests or Mann–Whitney tests. Data are presented as the mean ± SEM; ns = not significant; * *p* < 0.05; ** *p* < 0.01; *** *p* < 0.001; **** *p* < 0.0001.

**Figure 4 cells-11-01094-f004:**
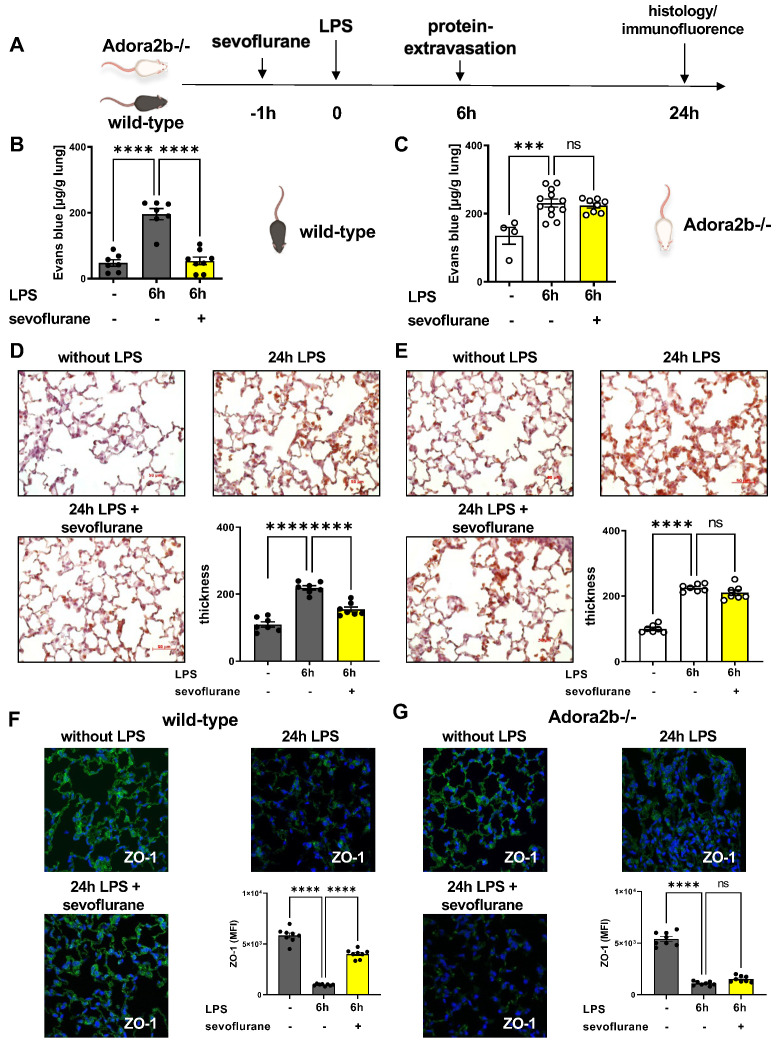
Impact of sevoflurane on microvascular permeability of wild-type and adenosine A2B receptor−/− (Adora2b−/−) animals. (**A**) Six and 24 h after lipopolysaccharide (LPS) stimulation and sevoflurane treatment, we evaluated the protein extravasation, and the pulmonary tissue thickness by immunohistochemistry. (**B**) The impact of sevoflurane on the LPS-induced Evans blue extravasation in wild-type, and (**C**) Adora2b−/− mice (*n* = 6–8). (**D**) Histological slides of the pulmonary tissue with or without LPS exposure of wild-type and (**E**) Adora2b−/− animals (*n* = 7–8). Evaluation of the septal thickness at indicated conditions. (**F**) The LPS-related expression of the tight junction protein zonula occludin-1 (ZO-1; green) in the lung tissue of wild-type and (**G**) Adora2b−/− mice was determined by immunofluorescence (mean fluorescence intensity; MFI) (original magnification ×63; one representative image of three independent experiments is shown; *n* = 3). DAPI (blue) was used as a nuclear marker. Fluorescence intensities of ZO-1 with or without LPS exposure were measured by using ImageJ (*n* = 6–8). Statistical analyses were performed by one-way ANOVA + Bonferroni correction. Data are presented as mean ± SEM; ns = not significant; *** *p* < 0.001; **** *p* < 0.0001.

**Figure 5 cells-11-01094-f005:**
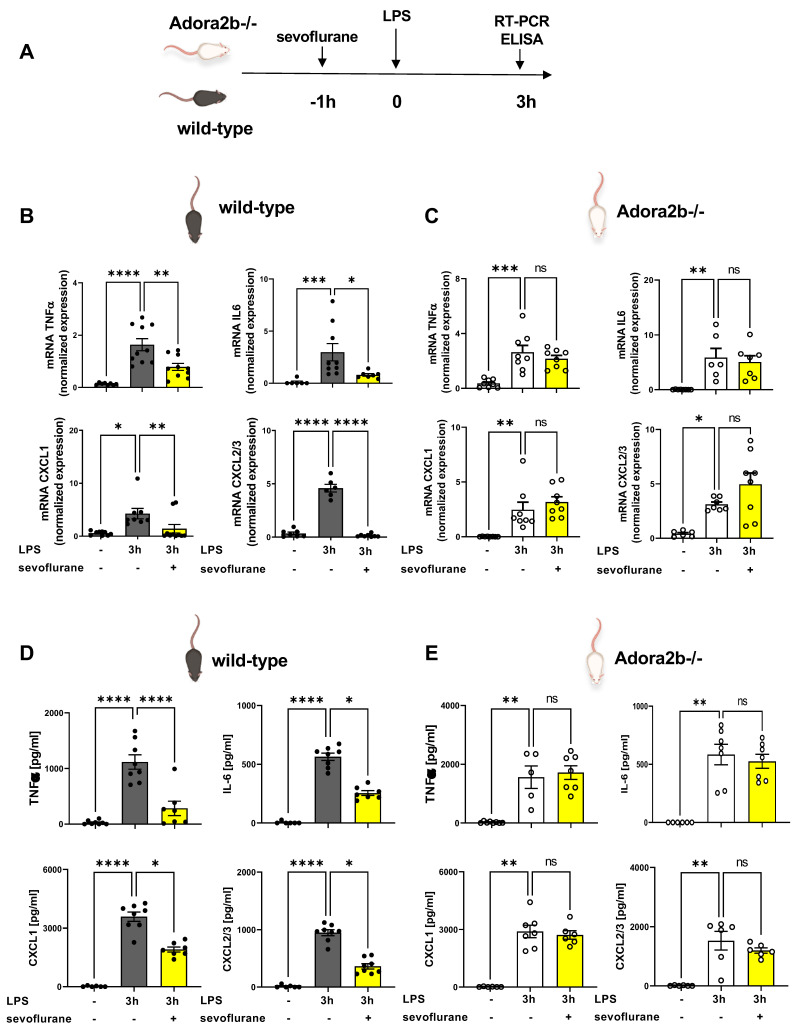
Influence of sevoflurane on the release of cytokines and chemokines in the bronchoalveolar lavage. (**A**) Wild-type and adenosine A2B receptor−/− (Adora2b−/−) mice were treated with sevoflurane. Three hours after LPS exposure, the gene expression (*n* = 6–10) and the release of inflammatory mediators such as tumor necrosis factor (TNF) α, interleukin (IL) 6, CXCL1, and CXCL2/3 were determined by real-time polymerase chain reaction and enzyme-linked immunosorbent assay from the pulmonary tissue and bronchoalveolar lavage of (**B**,**D**) wild-type and (**C**,**E**) Adora2b−/− animals (*n* = 6–8). Statistical analyses were performed by one-way ANOVA + Bonferroni correction or Kruskal-Wallis test. Data are presented as mean ± SEM; ns = not significant * *p* < 0.05; ** *p* < 0.01; *** *p* < 0.001; **** *p* < 0.0001.

**Figure 6 cells-11-01094-f006:**
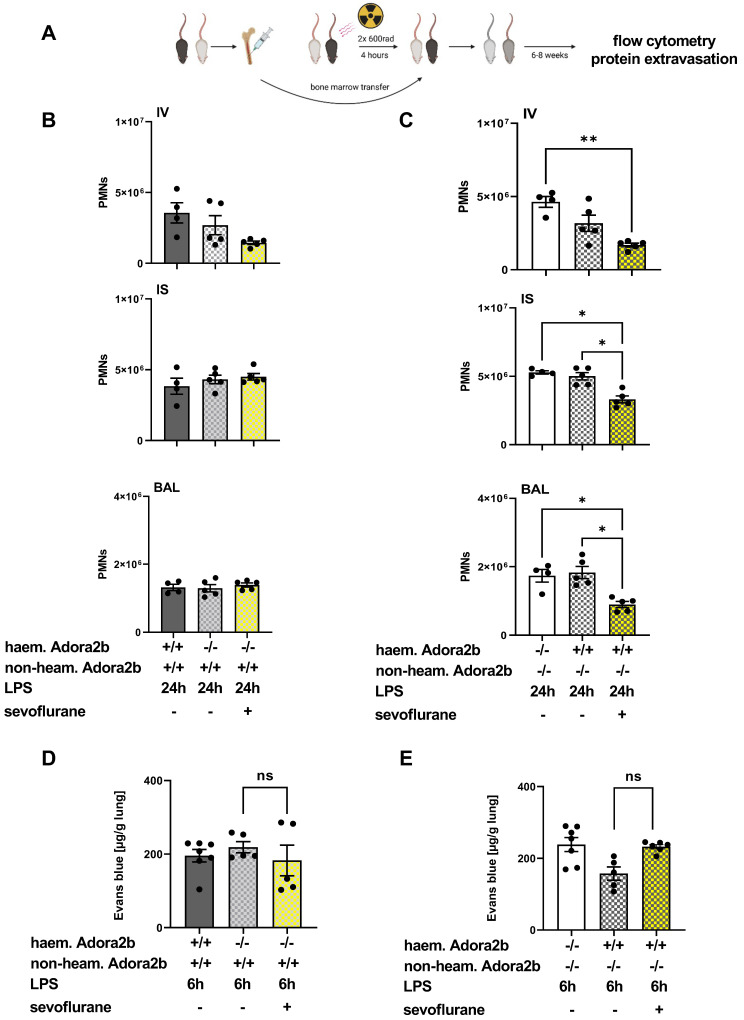
Contribution of sevoflurane on hematopoietic and non-hematopoietic cells to lipopolysaccharide (LPS)-related polymorphonuclear neutrophil (PMN) migration and microvascular permeability. (**A**) We generated chimeric mice by transferring bone marrow (BM) between wild-type (Adora2b+/+) and adenosine A2B receptor−/− (Adora2b−/−) animals to characterize the cell types, which contributed to the sevoflurane-related protective effects. Bone marrow transfers were performed: (1) BM from Adora2b−/− mice were injected into wild-type recipients (chimeric mice: Adora2b−/− hematopoietic cells/Adora2b+/+ non-hematopoietic cells), and (2) BM from wild-type (Adora2b+/+) animals were injected into Adora2b−/− mice (chimeric mice: Adora2b+/+—hematopoietic cells/Adora2b−/−—non-hematopoietic cells). (**B**) PMN counts from wild-type (Adora2b+/+) and chimeric (Adora2b−/−—hematopoietic cells/Adora2b+/+—non-hematopoietic cells) animals in the intravascular space (IV), pulmonary interstitium (IS), and bronchoalveolar lavage (BAL) were assessed by flow cytometry 24 h after LPS exposure. (**C**) Furthermore, the PMN migration from Adora2b−/− and the corresponding chimeric (Adora2b+/+—hematopoietic cells/Adora2b−/− non-hematopoietic cells) mice were quantified by flow cytometry (*n* = 4–5). (**D**) Extravasation of Evans blue was determined six hours after LPS exposure in the lung tissue. Further, we evaluated the impact of sevoflurane on Evans blue extravasation in wild-type (Adora2b+/+), Adora2b−/− mice, and Adora2b-tissue chimeric (Adora2b−/−—hematopoietic cells/Adora2b+/+ non-hematopoietic cells), respectively (**E**) Adora2b-blood chimeric animals (Adora2b+/+—hematopoietic cells/Adora2b−/− non-hematopoietic cells) (*n* = 4–8). Statistical analyses were performed by one-way ANOVA + Bonferroni correction or Kruskal-Wallis test. Data are presented as mean ± SEM; ns = not significant; * *p* < 0.05; ** *p* < 0.01.

**Figure 7 cells-11-01094-f007:**
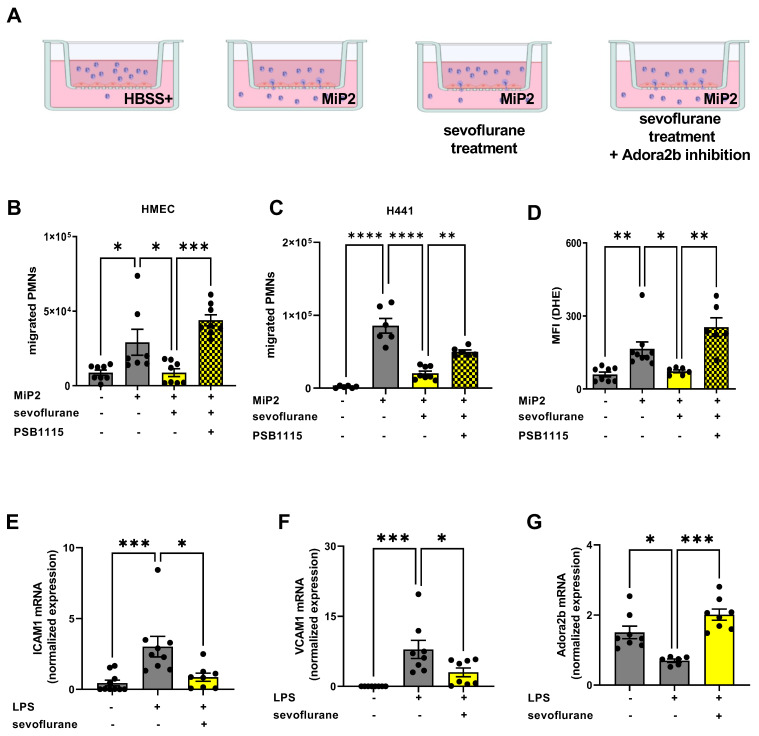
Impact of sevoflurane on the migratory behavior of human polymorphonuclear neutrophils (PMNs) in vitro. (**A**) Model of adhesion and transmigration assay to evaluate the neutrophilic diapedesis and to determine the impact of sevoflurane on PMN migration. (**B**) Human PMNs from healthy volunteers were isolated and treated by sevoflurane, respectively, sevoflurane and a specific adenosine A2B receptor (Adora2b) antagonist (PSB1115). The migration through a monolayer of human endothelial (HMEC) and (**C**) epithelial monolayer (H441) was initiated by the potent chemoattractant macrophage inflammatory protein-2 (MIP-2). (**D**) The impact of sevoflurane on reactive oxygen species activity of human PMNs was assessed by the quantification of dihydroethidium (DHE) activity (mean fluorescence intensity; MFI) (*n* = 6–8). (**E**) The gene expression of the human adhesion-related molecules, intercellular adhesion molecule 1 (ICAM1), (**F**) vascular cell adhesion molecule 1 (VCAM1), and (**G**) Adora2b expression were detected by RT-PCR (*n* = 6–10). Statistical analyses were one-way ANOVA + Bonferroni correction or Kruskal-Wallis test. Data are presented as mean ± SEM. * *p* < 0.05; ** *p* < 0.01; *** *p* < 0.001, and **** *p* < 0.0001.

**Figure 8 cells-11-01094-f008:**
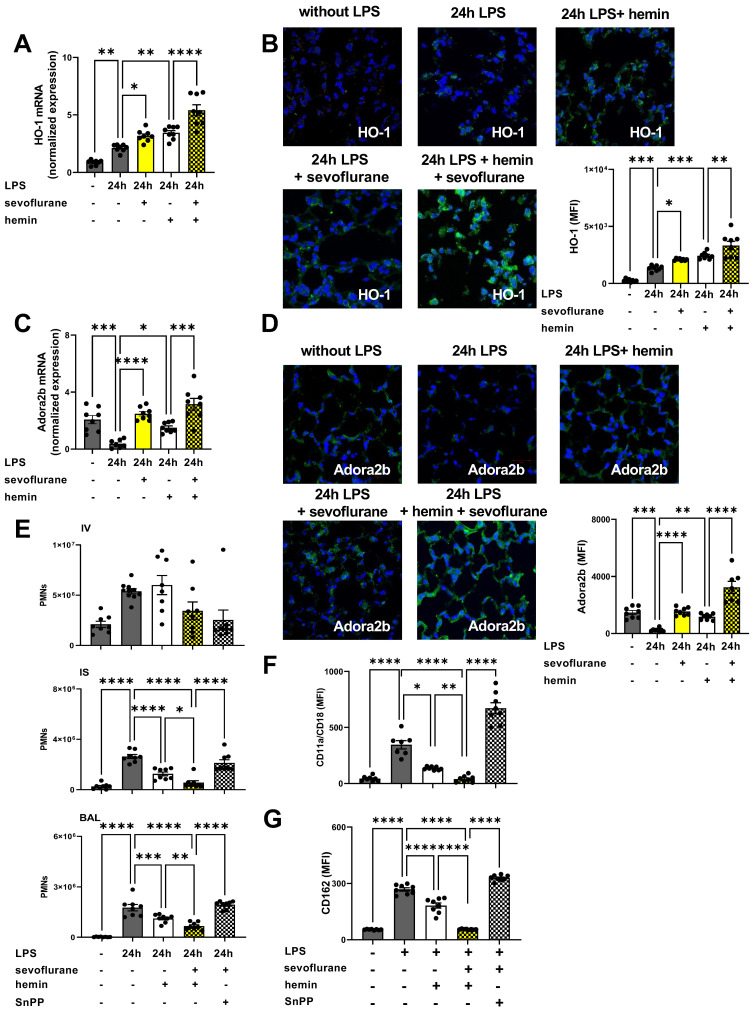
The impact of sevoflurane on heme oxygenase-1 (HO-1) modulation during acute pulmonary inflammation. (**A**) Gene, and (**B**) protein expression of HO-1 was evaluated in the pulmonary tissue three and 24 h after lipopolysaccharide (LPS) exposure. The effects of sevoflurane on HO-1 gene expression were examined by real-time polymerase chain reaction (RT-PCR). HO-1 protein expression (green) in the lung tissue of wild-type animals was evaluated by immunofluorescence staining at indicated conditions (original magnification ×63; one representative image of three independent experiments is shown; *n* = 3). DAPI (blue) was used as nuclear marker. Mean fluorescence intensity (MFI) of HO-1 with or without LPS exposure was measured by using ImageJ (*n* = 6–8). (**C**) Adenosine receptor A2B (Adora2b) gene expression was determined in the lung tissue after sevoflurane treatment, respectively, sevoflurane and hemin combination three hours after LPS exposure (*n* = 6–8). (**D**) Immunofluorescence staining of Adora2b surface expression (green) in lung tissue of wild-type animals was evaluated at indicated conditions (original magnification ×63; one representative image of three independent experiments is shown; *n* = 3). DAPI (blue) was used as a nuclear marker. MFI of Adora2b surface expression was measured at indicated conditions by using ImageJ (*n* = 6–8). (**E**) The impact of HO-1 activation or inhibition on sevoflurane effects on polymorphonuclear neutrophil (PMN) migration 24 h after lipopolysaccharide (LPS) stimulation was detected (*n* = 8). (**F**) Surface expression of CD11a/CD18 (also known as lymphocyte function-associated antigen-1; LFA-1), and (**G**) CD162 (also called P-selectin glycoprotein 1; PSGL1) was evaluated on human PMNs after LPS exposure. The impact of sevoflurane and heme oxygenase modulation on the presence of CD11a/CD18 and CD162 (mean fluorescence intensity; MFI) on human PMNs was assessed by flow cytometry (*n* = 6–8). Statistical analyses were performed by one-way ANOVA + Bonferroni correction or Kruskal-Wallis test. Data are presented as mean ± SEM; * *p* < 0.05; ** *p* < 0.01; *** *p* < 0.001; **** *p* < 0.0001.

**Figure 9 cells-11-01094-f009:**
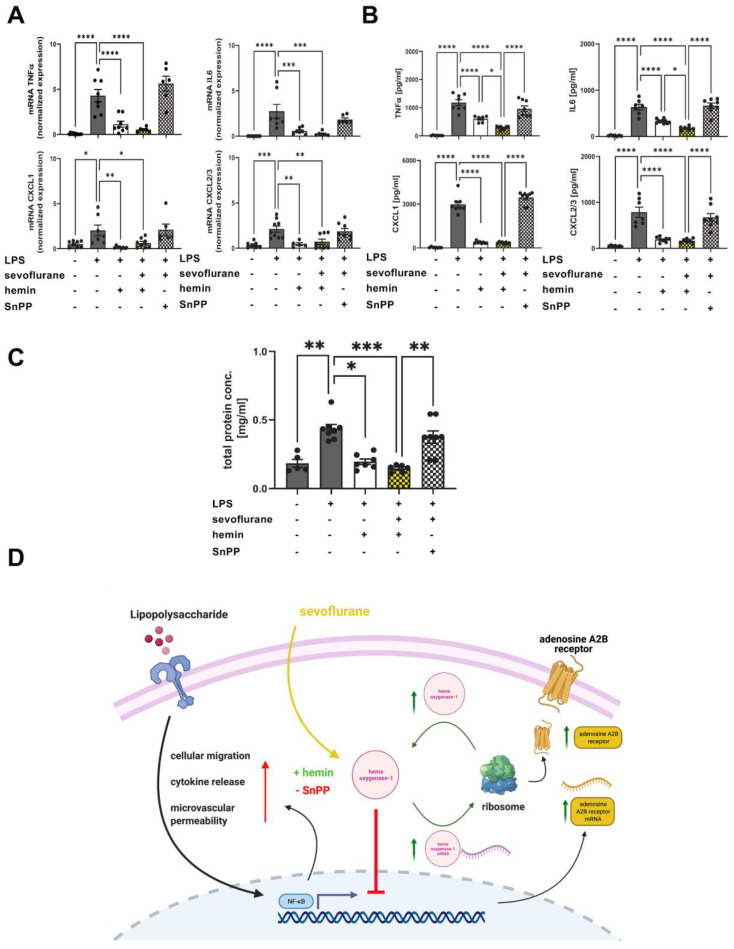
The additive effects of sevoflurane and heme oxygenase-1 (HO-1) activator on the release of inflammatory cytokines and microvascular permeability during acute lung injury. (**A**) The gene expression of tumor necrosis factor (TNF) α, interleukin (IL) 6, CXCL1, and CXCL2/3 in the lung tissue under the influence of HO-1 modulation during acute LPS-induced pulmonary inflammation was determined by real-time-polymerase chain reaction (RT-PCR) (*n* = 6–12). (**B**) The TNFα, IL6, CXCL1, and CXCL2/3 release in the alveolar space was assessed after LPS exposure at indicated conditions by using enzyme-linked immunosorbent assay (ELISA) (*n* = 6–8). (**C**) Additive effects of hemin activation on sevoflurane treatment during microvascular permeability were investigated. Evans blue extravasation in the pulmonary tissue were evaluated six hours after LPS stimulation at indicated conditions (*n* = 6–8). (**D**) Schematic overview of sevoflurane effects on heme oxygenase-1 and adenosine A2B receptor expression during acute pulmonary inflammation. Images were created with BioRender. Statistical analyses were performed by one-way ANOVA + Bonferroni correction or Kruskal-Wallis test. Data are presented as mean ± SEM; * *p* < 0.05; ** *p* < 0.01; *** *p* < 0.001; **** *p* < 0.0001.

## Data Availability

The datasets generated during and/or analyzed during the current study are available from the corresponding author on reasonable request.

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
