# Peer review of "Sevoflurane Dampens Acute Pulmonary Inflammation via the Adenosine Receptor A2B and Heme Oxygenase-1"

_cells, 2022, doi:10.3390/cells11071094_

Round 1

Reviewer 1 Report

Some of the descriptions of the content are confusing. For instance, 57 and 82 lines in page 2,  'respectively' seems inappropriate. In figure 1A, where is LPS alone as a control? 

Reviewer 2 Report

In this manuscript, authors observed that sevoflurane exhibits protective effects by modulating the expression of various adhesion molecules on PMNs during LPS-induced acute lung injury. Additionally, sevoflurane enhanced the microvascular permeability and reduced the release of inflammatory cytokines, whereas the protective effects were associated with Adora2b. Overall, the manuscript present comprehensive data to address possible mechanisms of sevoflurane actions. The data are interesting and important.  

Major:

1- “Sevoflurane was applied one hour before (pre) LPS inhalation, immediately after LPS (sim), or three hours (post) after LPS stimulation”. How did authors determine the time timepoints of sevoflurane treatment?

2- “The cellular response by adenosine occurs through the four adenosine receptors A1 (Adora1), Adora2a, Adora2b, and Adora3”. There are four receptors, the author should measure the expression levels of the other two receptors.

3- Does sevoflurane impact CD39, CD73 expression or adenosine concentration in BAL during LPS-induced lung inflammation?

4- Figure legends for Fig. 6 should be clear. Although authors gave the following information: 1) BM from Adora2b-/- mice were injected into wild-type recipients (chimeric mice: Adora2b-/- hemato- poietic cells/ wild-type - tissue) and 2) BM from wild-type animals were injected into Adora2b-/- mice (chimeric mice: wild-type - hematopoietic cells/ Adora2b-/- - tissue). But it is still difficult to tell which data are 1) or 2)?

5- Figure 9D- Schematic overview of sevoflurane effects on heme 638 oxygenase-1 and adenosine A2B receptor expression during acute pulmonary inflammation. It seems that authors did not have much information on ribosome. Should base on the work of this study. Fonts are too small for some.

Minor:

1- There are some mistakes in the manuscript, the author should examine the manuscript carefully.

2- in vivo, It should be italic in vivo.

3- Line 413: occluding should be occludin.

Round 2

Reviewer 2 Report

Authors have address all of my concerns well. Authors have supplied the data of the gene levels of Adora1, Adora3, CD39, CD73 after LPS in presence or absence of sevoflurane. These data must be included in the manuscript in the section of either Figures or supplemental Figures.
